# Biochemical and Cellular Characterization of New Radio-Resistant Cell Lines Reveals a Role of Natural Flavonoids to Bypass Senescence

**DOI:** 10.3390/ijms23010301

**Published:** 2021-12-28

**Authors:** Maria Russo, Carmela Spagnuolo, Stefania Moccia, Idolo Tedesco, Fabio Lauria, Gian Luigi Russo

**Affiliations:** National Research Council, Institute of Food Sciences, 83100 Avellino, Italy; carmela.spagnuolo@isa.cnr.it (C.S.); stefania.moccia@isa.cnr.it (S.M.); idolo.tedesco@isa.cnr.it (I.T.); fabio.lauria@isa.cnr.it (F.L.)

**Keywords:** γ radiation resistance, cancer therapy, therapy-induced senescence, senolytics, flavonoids, BH3 mimetics

## Abstract

Cancer is one of the main causes of death worldwide, and, among the most frequent cancer types, osteosarcoma accounts for 56% of bone neoplasms observed in children and colorectal cancer for 10.2% of tumors diagnosed in the adult population. A common and frequent hurdle in cancer treatment is the emergence of resistance to chemo- and radiotherapy whose biological causes are largely unknown. In the present work, human osteosarcoma (SAOS) and colorectal adenocarcinoma (HT29) cell lines were γ-irradiated at doses mimicking the sub-lethal irradiation in clinical settings to obtain two radio-resistant cellular sub-populations named SAOS400 and HT500, respectively. Since “therapy-induced senescence” (TIS) is often associated with tumor response to radiotherapy in cancer cells, we measured specific cellular and biochemical markers of senescence in SAOS400 and HT500 cells. In detail, both cell lines were characterized by a higher level of expression of cyclin-dependent kinase inhibitors p16^INK4^ and p21^CIP1^ and increased positivity to SAβ-gal (senescence-associated β-galactosidase) with respect to parental cells. Moreover, the intracellular levels of reactive oxygen species in the resistant cells were significantly lower compared to the parental counterparts. Subsequently, we demonstrated that senolytic agents were able to sensitize SAOS400 and HT500 to cell death induced by γ-irradiation. Employing two natural flavonoids, fisetin and quercetin, and a BH3-mimetic, ABT-263/navitoclax, we observed that their association with γ-irradiation significantly reduced the expression of p16^INK4^, p21^CIP1^ and synergistically (combination index < 1) increased cell death compared to radiation mono-alone treatments. The present results reinforce the potential role of senolytics as adjuvant agents in cancer therapy.

## 1. Introduction

According to epidemiological data from the World Health Organization (WHO), by 2040, the number of new cancer cases per year is expected to rise to 29.5 million and the number of cancer-related deaths to 16.4 million (Global Burden of Disease Cancer 2019). This prediction can partially be explained by the slow progress in drug discovery and application of personalized treatments in poor and low-income countries and, in part, by the increased elderly population in Western countries [1]. Besides these sociological aspects, the roots of cancer resistance are extremely complex. It has been hypothesized that cancer cells, after surgical resection or after surviving chemo- or radiotherapy, become “dormant”, but metabolically active for a relatively long time before initiating re-growth [2]. The work by Kreso et al. [3] analyzed intra-tumor heterogeneity focusing on the biological differences between malignant cells that originated within the same tumor. Treatment of xenograft-bearing mice with oxaliplatin, a chemotherapeutic drug commonly used in colorectal cancer, preferentially eliminated persistent clones and increased the proportion of clones that were initially below the detection threshold. The authors concluded that chemotherapy promotes the dominance of previously minor or dormant lineages in the same tumor. On the same issue, Perego et al. [2] demonstrated in lung and ovarian cancers that the re-activation of dormant tumor cells was largely responsible for metastasis and cancer mortality.

Ionizing radiations (IR), or, as defined in clinical oncology, low-linear energy transfer (LET)-radiation are often used against some inoperable or chemo-resistant cancers (osteosarcoma, colorectal, breast and prostate cancer, some leukemia, and lymphomas) [4,5,6,7]. LET radiation induces about 70% of DNA damage by generating hydroxyl radical (^•^OH), superoxide anion (O_2_^−^), and hydrogen peroxide (H_2_O_2_) due to the chemical decomposition of the H_2_O molecule or water radiolysis [8]. The oxidative stress leads to the subsequent guanidine oxidation (8-oxoGua) or cytosine cleavage [7,9] and DNA double strand break [10]. Clinical evidence revealed that IR, failing to bypass the threshold necessary to trigger apoptosis, can favor the repopulation of tumor cells, stimulating the process known as “therapy-induced senescence” (TIS) on the sub-group of death-resistant cancer cells [11,12,13]. Senescence is a fundamental biological process in the regulation of cellular homeostasis and is often associated with the response of cancer cells to chemo- and radiotherapy [10]. It is related to DNA damage induced by genotoxic or mitogenic stress and cell cycle control [14]. Senescence was described in 1961 by Hayflick and Moorehead [15], and it is recognized as one of the hallmarks of aging and cancer [16]. It refers to a cell condition involving an irreversible replicative arrest, sustained viability, and resistance to apoptosis. While apoptosis is tumor suppressive and always related to cell death, recent studies revealed that senescence, similarly to autophagy, emerges later in time and is linked to cancer resistance [17,18,19,20]. The increased metabolic and inflammatory activity known as SASP (senescence-associated secretory phenotype) can contribute to senescence-related inflammation [21]. Understanding the pleiotropic roles of senescence associated with SASP is even more complex in cancer biology, wherein stress response to chemotherapy and radiotherapy can induce TIS [22]. The biochemical pathways linking TIS to genotoxic stress and the cell cycle have been studied since the nineties. These pathways involve changes in the gene expression of known cell-cycle inhibitors or activators, including the CDK (cyclin-dependent kinase) inhibitor p21^CIP1^ and p16^INK4^, a component of retinoblastoma (RB) tumor suppressor signalling cascades [23,24]. A solid amount of literature indicates that p53 causes growth arrest after DNA damage induced by γ−rays through transcriptional activation of p21^CIP1^ [25]. Mechanistically, there are multiple factors responsible for cellular senescence, both intrinsic or environmental, e.g., DNA damage, telomere shortening after each cell division, proteolytic stress due to an alteration in autophagy process, and reactive metabolites such as reactive oxygen species (ROS) [11,16]. The biochemical link between changes in cellular redox status and senescence was demonstrated in different contexts, such as oncogene-induced senescence (OIS), wherein RAS activation was linked to the upregulation of NADH oxidase subunits Nox1 and Nox2, the main regulators of ROS at the plasma membrane level [26]. Hubackova et al. [27] clearly indicated a mechanistic link between SASP cellular redox status and DNA damage signaling. They found that media conditioned by cells undergoing any of the three main forms of senescence (replicative, OIS, and therapy-induced) contained high levels of IL1, IL6, and TGFβ and were capable of inducing an ROS-mediated DNA damage response (DDR) in normal bystander cells through the JAK/STAT, TGFβ/SMAD, and IL1/NF-κB signaling pathways. 

Recent studies identified a new class of drugs called “senolytics” or “senotherapeutics” for their ability to selectively eliminate senescent cells or reduce SASP in different aging-related diseases (kidney diabetic disease and pulmonary idiopathic fibrosis) [28,29]. Senolytic drugs include BH3-mimetics, such as ABT-263/navitoclax, some tyrosine-kinase inhibitors, e.g., desatinib, and natural molecules belonging to the class of flavonoids, such as quercetin and fisetin [21,30]. Although the specificity and sensitivity of senolytics are not completely known, they have been proposed as a potential adjuvant approach to improve the response to those cancer therapies that induce senescence [31,32]. Several preclinical studies are reporting promising results [30]. As an example, in soft-tissue sarcoma cells that, when irradiated, undergo TIS, the addition of senolytic agents, venetoclax (ABT-199) or navitoclax in combination with IR induced rapid apoptotic cell death [33]. 

The present work aims to investigate the capacity of different senolytics, including quercetin, fisetin, and ABT-263, to increase IR sensitivity in two radio-resistant cellular sub-populations derived from human osteosarcoma and colon adenocarcinoma cell lines.

## 2. Results

### 2.1. Development of Radio-Resistant Sub-Populations of SAOS and HT29 Cells

To establish an in vitro model of cancer cells that can escape from radiotherapy-induced cytotoxicity, we selected SAOS and HT29 cell lines derived from human osteosarcoma and colon adenocarcinoma, respectively, two tumors often characterized by high resistance to chemo- and radiotherapy [34,35].

Cells (1 × 10^6^) were irradiated with sub-lethal doses of γ-rays using two cycles (the first at time = 0 and the second after 24 h) of 4 Gy (SAOS cell line) or 5 Gy (HT29 cell line). These doses were chosen to avoid massive cell death [36] and to mimic the “fractionated doses” of radiation used in clinical settings, e.g., 1.8–2 Gy in 4–5 cycles [37] (Figure 1). In the following three weeks after irradiation, SAOS and HT29 cells evidenced an extensive cell death (>60–80%) verified by Trypan blue exclusion dye count (data not shown). Starting from week 4, a few surviving colonies reinstated their growth, reaching standard levels of cell viability (>90%). We named SAOS400 and HT500, the sub-populations derived from parental SAOS and HT29 cell lines, respectively (Figure 1). To avoid changes in cellular and biochemical parameters due to long permanence in culture, in all experiments, SAOS400 and HT500 cells were used at the same early passages (<10).

Given that SAOS400 and HT500 cells showed specific proliferative kinetics, with a doubling time of about 48 h for SAOS400 and 20–24 h for HT500 (data not shown), cell viability was measured at different times from IR (Figure 2). As expected, SAOS400 and HT500 were significantly more resistant to a new burst of irradiation compared to parental cells (Figure 2a,c). In fact, the calculated EC50 (dose of IR required to reduce cell viability of 50%) was 8.4 Gy for SAOS400 cells with respect to 4.8 Gy in parental SAOS cells after 120 h from IR. Diversely, EC50 in HT500 cells was 50.6 Gy compared to 20.74 Gy in parental HT29 cells after 72 h from IR. Microphotographs of SAOS and SAOS400 cells stained with Crystal Violet revealed an enlarged and flattened cell morphology in both cell types after IR (red arrows in Figure 2b), although it was more abundant in SAOS400 cells. In HT29, and even more in HT500 cells, an altered nuclear size and/or multinucleated cells were detected by Cy-Quant staining after IR (red arrows in Figure 2d). In order to compare the proliferative ability and verify cellular recovery expressed as the surviving fraction (S.F.) after the irradiation of SAOS400 and HT500 cells, we performed colony-forming assays that lasted 12 days following the treatment, with different doses of γ-ray (5 and 10 Gy), starting from 1000 cells/well. As reported in Figure 3, the S.F. values indicated that a significantly higher number of colonies were formed in irradiated SAOS400 and HT500 cells compared to their parental cell lines. This assay also confirmed the higher resistance to IR of HT500 with respect to SAOS400 cells that can be explained evoking the different genetic backgrounds, the former being p53-mutated and the latter p53-null.

### 2.2. Cytostatic Response of SAOS400 and HT500 Cells to IR

The effects of irradiation on cell growth and cell death in SAOS400 and HT500 cells were investigated following irradiation with 5 Gy and monitoring cell viability at different times using Trypan-Blue exclusion dye. Cell count showed that cells remained viable, but the IR treatment induced a significant growth arrest in the irradiated SAOS400 and HT500 cells compared to their controls (Figure 4a,b). To confirm the growth arrest, the cell cycle distribution was analyzed. For both cell types, a double block was evidenced with an accumulation of cells in S and G2/M phases (64.47 and 21.8%, respectively) in irradiated SAOS400 cells (Figure 4c). Similarly, after irradiation, S and G2/M cells increased in HT500 cells to 46.20 and 29.12% (Figure 4d). Both cell lines showed no signs of apoptosis (data not shown).

### 2.3. Measurement of ROS and GSH Levels in SAOS400 and HT500 Cell Lines

The intracellular redox state was evaluated after IR in SAOS400 and HT500 cells and compared to their parental counterparts using two different probes: CM-DCFDA, specific for intracellular peroxides, and DHE, which detects superoxide anion (O_2_^−^). Baseline levels of ROS were similar in parental and radio-resistant cells; therefore, we investigated if IR was associated with changes in their redox status. Figure 5a shows that, following IR, peroxides production in SAOS400 cells was 20–30% lower within 5–15 from treatment with respect to parental SAOS. Similarly, HT500 produced 80–90% less intracellular peroxide compared to HT29 cells in the same experimental setting (Figure 5c). The trend was superimposable for superoxide production. Fluorescence of DHE probe recorded from 5 to 15 min after IR showed a significantly lower production of O_2_^−^ in resistant sub-populations. We measured <40% superoxide production in SAOS400 cells (Figure 5b) and <15% in HT500 cells (Figure 5d) compared to their respective parental cell lines. We also measured the levels of GSH, a critical cellular reducing agent involved in chemotherapy and radiotherapy resistance [38]. Figure 5e shows that GSH levels significantly increased (>20–25%) after 2 h from IR in both SAOS400 and HT500 cell lines. Interestingly, no GSH increase was observed in parental cells after 2 h from IR (data not shown).

### 2.4. Evaluation of Senescence Induced by γ-rays in SAOS400 and HT500 Cells

Growth arrest associated with senescence is a common response to repair DNA damage and, at a later time, can generate IR resistance in cancer cells [18,39]. To verify if γ-ray treatment induced senescence in normal and resistant sub-populations of SAOS and HT29 cells, different specific senescence markers were assessed. Figure 6a,b clearly reports that γ-ray treatment increased SA-βGal positivity by about 1.4-fold in SAOS cells with respect to untreated controls (black bars in Figure 6a). On the contrary, at baseline, SAOS400 presented significantly high levels of SA-βGal positivity with respect to the parental cells (about 4-fold); therefore, no significant difference in SA-βGal positivity was detected in SAOS400 cells after irradiation (compare the two red bars in Figure 6a). The high basal level of senescence in SAOS400 cells was also confirmed by C_12_FDG fluorescence staining, used to assess βGal enzymatic activity in live cells (Figure 6c,d) and further corroborated by the increased expression (about 2 fold) of p16^INK4^ in SAOS400 cells (Figure 6e). Similarly, in the HT29 cell line, a 3-fold increase of SA-βGal positivity was measured in irradiated cells with respect to untreated cells (Figure 7a,b). As for SAOS400 cells, basal HT500 cells also presented about 2.5-fold higher levels of SA-βGal positivity compared to their parental cells. 

However, differently than SAOS400, after irradiation, SA-βGal positivity significantly increased in HT500 cells (about 64%) with respect to untreated HT500 cells. Although less evident than in SAOS vs. SAOS400 cells, C_12_FDG fluorescence staining also confirmed that in the HT500 cell line the basal level of senescence was significantly higher compared to parental HT29 cells (about 31%; Figure 7c,d). This feature found a further validation in the expression of p16^INK4^ that was about 2-fold higher in HT500 compared to the parental cell line (Figure 7e). Being that HT29 cells p53-mutated while SAOS cells are p53-null, we measured in the former the change in the expression of p21^CIP1^ that resulted in over-expression (about 6-fold; Figure 7f). It is interesting to mention that these two senescence markers, p16^INK4^ and p21^CIP1^, showed time-dependent profiles that reflect the tumor plasticity and the presence of a heterogeneous population after IR recovery, as described in other cellular models [40].

### 2.5. Effect of Different Senolytics in Restoring IR Sensitivity in SAOS400 and HT500 Cell Lines

Selected senolytic agents were employed to verify the reduction of radio-resistant senescent cells and restore IR sensitivity in SAOS400 and HT500 cells. To this purpose, we firstly verified the potential activity of fisetin (F; 3,3′,4′,7-tetrahydroxyflavone), a natural molecule belonging to the flavonoid family, one whose senolytic properties have already been described [41]. Figure 8a shows that adding F after 72 h from IR (10 Gy) and measuring cell viability after an additional 72 h, the combined treatment (F + IR) reduced cell viability by 23%, and this decrease was significant with respect to IR single treatment in SAOS400 cells. No significant differences were observed when the combined treatment was applied to the parental SAOS cells. Similarly, we observed a lower reduction in cell viability (8.2%) in the combined treatment F + IR in HT500 cells with respect to IR mono-treatment with no significant differences in parental HT29 cells (Figure 8b). It is also worthwhile to note the different effects of F in the single treatments, which was almost negligible in HT29 and HT500 cells (Figure 8b), but measurable (about 40% decrease on cell growth) in parental SAOS cells and, to a lesser extent, in SAOS400 (Figure 8a). Based on these observations, we performed a series of additional experiments changing the F concentrations applied and the times of treatments to better highlight the efficacy of the combined treatment, F + IR (data not shown). In Figure 8c,d, we show the results of the modified experimental setting wherein F was applied at 40 μM for 96 h soon after 10 Gy irradiation. In both SAOS400 and HT500 cell lines, a clear and significant increase in cell death was measured following the combined treatment. More in detail, the combination of F plus IR decreased cell viability by 50% compared to irradiated-only SAOS400 cells (Figure 8c), while for HT500 cells the association F plus IR lowered the percentage of viable cells by about 34% with respect to IR single treatment. In these experiments, daunorubicin (D), a well-known chemotherapeutic drug, was used as a positive control and, as expected, its cytotoxicity ranged between 70 and 80% in both cell lines.

To reinforce the observation that natural senolytics could restore sensitivity to radiotherapy, we used a different compound with similar properties, i.e., quercetin (Q; 3,3′,4′,5,7-pentahydroxyflavone) [28]. Figure 8e,f show that, when adding 25 μM Q after IR treatment (5 Gy), a significant reduction in cell viability of about 29% for SAOS400 and 24% for HT500 was observed with respect to IR treatment alone in both cellular models. Similar results were obtained in both radio-resistant cells after combined treatment with IR plus 50 μM Q (data not shown). To support the demonstration that senolytic agents, both natural and synthetic, were able to sensitize radio-resistant cancer cells, the combination index (C.I.) was calculated to verify if the combined treatment between Q plus IR was synergistic or additive in SAOS400 and in HT500 cells. The Chou-Talalay method was employed to determinate C.I. using data obtained from the performed cell viability assays. As a positive control, we also calculated the C.I. for the combined treatment γ-rays plus ABT-263/navitoclax a well-known senolytic drug [41,42]. According to the Chou-Talalay method, cell viability measured by Cy-Quant assay was used to calculate the fraction affected (f.a.) and the C.I. for different concentrations of selected compounds and IR treatment at no constant ratio in radio-resistant cells. Data in Table 1 indicate that the C.I. for all the combined treatments analyzed were <1, confirming the synergistic effects of γ-rays when combined with Q or ABT-263.

Finally, to confirm that the senolytic effects of Q and F was able to counteract the senescence phenotype associated with cell radio-resistance, we evaluated the expression of senescence markers in SAOS400 and HT500 cells treated with IR (10 Gy), F, Q (both tested at 40 μM concentration), and their combinations. As shown in Figure 9 a clear reduction of p16^INK4^ was observed in the combined treatment F plus IR and Q plus IR compared to mono-treatments in SAOS400 (Figure 9a) and HT500 (Figure 9b) cells, confirming that the reduction in cell viability in both radio-resistant cell lines coincided with a parallel decrease of the senescence markers. Being HT500 cell line p53-mutated, we also evaluated the levels of p21^CIP1^, whose expression increased after treatment with IR, F, and Q in single treatments, while in both the combined treatments, i.e., F plus IR and Q plus IR, a significant reduction was measured (Figure 9c).

## 3. Discussion

Targeting small sub-populations of tumor cells resistant to chemo- or radiotherapy is considered a promising strategy for more effective therapeutic responses in patients [13,44]. The radio-resistant sub-populations obtained in this study were characterized by an “enrichment” in senescent cells, a phenotype well-studied in colorectal, non-small lung cancer, and breast tumor cell lines [40]. These cells were able to form a tumor mass when implanted both in immunocompetent and nude mice, clearly demonstrating that a senescent-like phenotype can contribute to the “awakening” of dormant cancer cells and, later in time, to disease recurrence [40]. In agreement with these concepts, the two radio-resistant cellular sub-populations investigated in the present study and obtained from the parental SAOS and HT29 cell lines maintained their proliferative capacity and showed features of “senescence-like arrest” [45]. The proliferative phenotype may appear to be a paradox, considering that senescence was historically defined as an irreversible process characterized by a durable and prolonged G1 growth arrest [15,16,24]. However, evidence exists on specific cellular models supporting the hypothesis that, if senescence is linked to an appropriate genetic background, it is not necessarily an irreversible process [46]. In fact, Dirac et al. showed the reversal of senescence in mouse embryonic fibroblasts (MEF) through the suppression of p53 with consequent immortalization and the loss of expression of senescence markers [47]. A similar process was described by others in senescent human fibroblasts and mammalian epithelial cells in a genetic background dependent on different tumor suppressors (p53, p16^INK4^, RB) or oncogenes (Ha-RAS^Val12^) [48]. These studies indicated that senescence due to telomere dysfunction is reversible and is maintained primarily by p53. However, p16I^NK4^ and RB provided a dominant second barrier to the unlimited growth of human cells because senescent cells with high levels of p16^INK4^ failed to proliferate upon p53 inactivation or RAS expression, but if RB was inactivated, they re-entered the cell cycle [48]. In the present study, we employed two well-established cancer cell lines with a documented highly modified genetic background. According to the ATCC catalogue and Sanger genomic database (https://cancer.sanger.ac.uk/cell_lines, accessed on 6 June 2021), SAOS cells presented homozygous deletions both in p53 and RB tumor suppressors; thus, this condition justified the observed phenotype in the SAOS400 sub-population, probably because the barriers opposing the “reversal of senescence” were lost. Similarly, the HT29 cell line presents genetic alterations and, in particular, the homozygous mutation in the DNA binding region of p53 resulting in a protein that is mutated and overexpressed [49]. This can partly explain the intrinsic higher resistance to the γ-rays of parental cell line HT29 compared to SAOS cells (Figure 2a,c). In fact, the p53 status may have an important role in modulating the IR response in osteosarcomas. This is supported by the observation that the U2OSs cell line, also derived from osteosarcoma and characterized by a wild-type p53, was more resistant to γ-rays than SAOS cells (data not shown) [50]. However, we noted that, while the SAOS400 cell line probably reached a plateau of senescence (Figure 6a), in HT500 cells this condition was not achieved (Figure 7a), leading to the hypothesis that additional regulatory pathways are active in HT29 and HT500 cells to evade IR-induced cell death, such as autophagy [51]. We are intensively working on this possibility that will be the focus of future works. SAOS400 and HT500 cells represent heterogeneous cellular populations; therefore, it is plausible that other additional phenotypes, such as protective autophagy and the “senescence-like state” associated with self-renewal potential, can explain the re-emergence of cancer cells after sub-lethal doses of γ-rays. Sabisz et al. [52] showed that non-small lung adenocarcinoma A549 cells, after treatment with genotoxic drugs, became permanently growth-arrested because of the so-called “pseudo-senescence”. However, a small fraction of drug-treated cells escaped pseudo-senescence, leading to the re-growth of the tumor cell population after drug treatment. The authors showed that this re-growth was associated with the presence of cancer stem cells (CSCs) in the lung tumor cell population [52]. In the present study, we did not investigate the presence of CSCs within the sub-populations of SAOS and HT29 cells selected after γ-ray irradiation, although their appearance and enrichment after drug treatments is a possible event, as reported by others [53,54]. For these reasons, we cannot exclude that “senescence associated-stemness” can be responsible for IR resistance in SAOS400 and HT500 cell lines.

We demonstrated that SAOS400 and HT500 cells showed a lower production of intracellular ROS compared to the respective parental counterparts (Figure 5). This observation can be explained considering that ROS are critical mediators in IR-induced cell death. In fact, Dihen et al. [55] showed that CSCs in radio-resistant human and murine breast tumors developed less DNA damage compared to normal stem cells. They also demonstrated that lower ROS levels in CSCs were associated with the increased expression of free radical scavenging systems such as glutamate-cysteine ligase (GCL) and HIF-1α. In particular, enzymatic antioxidant systems, such as those regulated by the stress-response transcription factors Nrf2, able to control specific ROS-mediated signaling pathways, can play an important role in the context of cancer resistance. Nrf2/Keap1 signaling governs the expression of the xCT (also called SLC7a11 or system xc–), involved in the transport of cysteine into the cell, along with GCL, which catalyzes the rate-limiting step in GSH biosynthesis [56]. Alternatively, a study on pancreatic cancer demonstrated that gemcitabine, a common chemotherapeutic drug, induced cytotoxicity through ROS generated by NADPH oxidase (NOX) and through the increase of p22 (phox) expression via NF-κB activation. As a feedback mechanism, the activation of Nrf2 stimulated the transcription of cytoprotective antioxidant genes, especially genes encoding enzymes that catalyze GSH production to reduce the increment of ROS as “an intrinsic resistance mechanism” [57]. Based on these data, we can interpret the results shown in Figure 5 reporting lower ROS levels after γ-ray treatment in radio-resistant cells and the parallel increase in intracellular GSH as being dependent upon the (over)-activation of Nrf2 and/or the (over)-expression xCT or protective antioxidant enzymes. Among these, putative candidates under scrutiny for future works are enzymes associated with GSH production (GCL) or NQO1 (NADPH quinone oxidase 1) linked to O_2_- scavenging.

We also observed that radio-resistant cells, after prolonged treatment with IR (96 h), were arrested in the G2/M (HT500 cells) and/or the G2/M and S phase (SAOS400 cells) of the cell division cycle (Figure 4). Although oxidative stress has been shown to induce senescence, it is difficult to correlate a very rapid event such as ROS increase, occurring within 30 min from the treatment, to dynamic events such as cell cycle arrest and senescence that take place after 72–96 h. Venkatachalam et al. [58] showed that, in rat myoblasts L6 and human retinal cells, RPE1-hTERT, a sub-lethal amount of exogenous H_2_O_2_, induces two waves of DNA damage. The authors noted that the first wave is rapid and transient while the second wave coincides with the cell transition from the S to the G2/M phases of the cell cycle. Finally, cells entered growth arrest accompanied by the acquisition of senescence-associated characteristics (SA-βGAL positivity), and this phenotype was independent of p53 status. This study not only goes in the direction of our results and opens the possible link between intracellular ROS production induced by γ-ray and the subsequent growth arrest in the G2/M and/or S phase but also explains the SA-βGAL positivity reported in Figure 6 and Figure 7 for SAOS400 and HT500 cells, respectively. However, as mentioned above, we cannot exclude the activation of other protective and dynamic cellular processes, such as protective or cytostatic forms of autophagy or autophagy-dependent senescence [59]. A recent study demonstrated that the selective autophagy of specific regulatory components coordinates the homeostatic state of senescence. In particular, the selective autophagy of Keap1, the cytoplasmic inhibitor of Nrf2, promoted redox homeostasis during senescence. Interestingly, these selective autophagy networks are clearly observed in vivo in human osteoarthritis [60].

Redox modulation can critically influence different pathways that fine-tunes oxidant mediated signals in cancer and normal cells [61], but the specific and selective elimination of senescent cells, responsible for γ-ray resistance using pro-oxidant/cytotoxic drugs, such as daunorubicin, is difficult. For this reason, a senolytic approach may be of therapeutic interest if aimed to the specific elimination of senescent cells [11,21]. Nonetheless, it is known that the effectiveness of senolytics is strictly dependent upon the time of administration [44]. In fact, in our study, we observed that SAOS400 and HT500 cells were more sensitive to the combined treatment of fisetin or quercetin plus γ-ray compared to parental cells when senolytics were added 72 h after irradiation (Figure 8). Interestingly, in HT500 cells, a clear cytotoxic effect was achieved when fisetin was added immediately after γ-ray treatment (Figure 8d) and, confirming the selectivity of several senolytic agents, the radio-sensitizing action of fisetin was absent in the same experimental condition in the parental HT29 cells (data not shown). These results confirm that natural senolytics, due to their well-known low stability in cell culture medium and poor intracellular bioavailability, are most effective when used in “a hit and run” modality, which, at least in vivo, can represent a procedure that avoids potential toxicity and off-target effects [44].

More intriguing is the senolytic molecular mechanism(s) of action of fisetin and quercetin in the parental and radio-resistant cells investigated in the present study (Figure 8 and Figure 9). In fact, these compounds can act directly or indirectly on multiple kinases associated with SASP and senescence or autophagy, such as PI_3_K/mTOR or MAPK pathways [62,63,64] overcoming the multiple and not fully explored resistance mechanisms emerging after γ-ray irradiation [65]. Lu et al. [66] showed that fisetin decreases the activities of CDK2 and CDK4 in HT29 cells by lowering the levels of cyclin E and D1 and increasing p21^CIP1^ expression [66]. In the same cell line, Kim et al. [67] demonstrated that quercetin favors cell cycle arrest in the G1 phase and up-regulates apoptosis-related proteins, such as p53 and p21^CIP1^. Our data confirmed the increased p21^CIP1^ expression in HT500 cells after fisetin and quercetin mono-treatments (Figure 9c). This observation can explain the “senolytic” effects of both compounds resulting in slowing down cell proliferation and/or inducing autophagy in the absence of cytotoxicity (no cell death measured until 96 h), as frequently observed after treatment of cancer cell lines with natural bioactive molecules [68]. Our data also coincide with the “pleiotropic nature” of flavonoids [64] and identify a “knowledge gap” in TIS and in the molecular mechanism of natural senolytics that deserves future investigations [44].

To summarize, the concept of a “one-two punch” cancer therapy [44] consisting of therapeutic interventions, such as radiotherapy, able to induce tumor cell senescence followed by selective clearance of senescent cells with senolytics, is in agreement with our results.

As depicted in Figure 10, radio-resistant cells, such as SAOS400 and HT500, represent a feasible model of TIS, a process, as we discussed, functionally heterogeneous and context-dependent (tissue of origin, time after insult, genetic background) [44] but always associated with an anti-apoptotic or cell-death-resistance phenotype. On the other side of the coin, TIS may display a therapeutic opportunity (first punch) because the clearance of senescent cells by senolytics can restore cell death sensitivity (second punch).

The use of senolytic drugs in clinics, if confirmed by appropriate trials, could result in an efficient alternative to spare delicate tissues from systemic cytotoxicity of γ-ray irradiation [28,29]. In fact, in vivo treatment with the senolytics cocktail, dasatinib plus quercetin, mitigates radiation ulcers representing a prevalent toxic side effect in patients receiving radiotherapy [69]. Further investigations are needed to define the features of radio-resistant cellular sub-populations in-depthand clarify how natural flavonoids can function as “senolytics adjuvants” in killing these cell types when associated with radiotherapy.

## 4. Materials and Methods

### 4.1. Cell Culture and γ-ray Treatment

The SAOS [70] cell line was a generous gift from Prof. A. Oliva (L. Vanvitelli University, Naples, Italy), while the HT29 [71] cell line was purchased from LGC Standards (Sesto San Giovanni, MIlan, Italy). Both cell lines were cultured in Dulbecco’s Modified Eagle’s Medium (DMEM; Euroclone, Milan, Italy) supplemented with 10% foetal bovine serum (FBS; Thermo-Fisher Scientific/Life Technologies, Milan, Italy) and added with 100 µg/mL penicillin/streptomycin, 2 mM L-glutamine, and 100 µM non-essential amino acids (Thermo-Fisher Scientific/Life Technologies, Milan, Italy) at 37 °C in a humidified 5% CO_2_ atmosphere. To avoid the “age effects”, cells were used at early passage (<10) in the obtained results. The SAOS and HT29 cells (1 × 10^6^) were irradiated with 4 and 5 Gy, respectively, at time 0 and after 24 h, using a Gammacell Elite 1000 instrument (MDS Nordion, Ottawa, ON, Canada) equipped with a radioactive source emitting γ rays (Cesium^137^emitting about 2.5 Gy/minute). Two sub-populations enriched in radiation-resistant cells were named SAOS400 and HT500, obtained by irradiating cells (1 × 10^6^ cells) with two cycles (the first at T = 0, the second after 24 h) of 4 and 5 Gy, respectively. Subsequently, cells were cultured for 4 weeks, allowing the expansion of the survived radio-resistant cells, which were trypsinized with Trypsin/EDTA solution (Euroclone) and counted with Trypan blue exclusion dye (Merck/Sigma, Milan, Italy) using an automatic cell counter (EveTM, NanoEnTek distributed by VWR, Milan, Italy). The obtained sub-populations enriched in radiation-resistant cells were named SAOS400 and HT500. For cell viability assays, cells (2 × 10^4^/mL) were irradiated with doses ranging from 2.5 Gy to 40 Gy. In the experiments with the senolytic drugs, cells were pre-irradiated (5 and 10 Gy) and subsequently treated with fisetin (F, 20–40 μM; Merck/Sigma, Milan, Italy), quercetin (Q, 25, 40, 50 μM; Merck/Sigma, Milan, Italy), ABT-263/navitoclax (ABT-263, 0.5–1.0 μM; ApexBio, Boston, MA, USA), or daunorubicin hydrochloride (a gift from Dr. Silvestro Volpe from S.G. Moscati Hospital, Avellino, Italy) for the indicated times (72–96 or 120 h). Two cell viability assays, Crystal Violet (Merck/Sigma, Milan, Italy) and Cy-Quant (Thermo-Fisher Scientific/Life Technologies, Milan, Italy), were performed as described [68,72]. Crystal Violet is based on the staining of attached, fixed cells with a dye, which binds to proteins and DNA. Briefly, cells were fixed with 10% formalin for 15 min and 0.1% Crystal Violet (*w/v*) was added for 30 min. After washing, cells were solubilized with 10% acetic acid, and absorbance was measured spectrophotometrically at 590 nm. CyQuant is a cell-permeant fluorescent DNA-binding dye (CyQuant nuclear stain) used in combination with a masking dye reagent (background suppressor) able to quantify cell proliferation and cytotoxicity. Briefly, the CyQuant mixture was added to the culture medium for 1 h at 37 °C, and fluorescence was measured at an excitation wavelength of 485 nm and emission of 530 nm in a multiplate reader (Synergy HT, BioTek, Milan, Italy). The two assays were performed simultaneously with cell counts (Trypan blue, Merck/Sigma, Milan, Italy) to confirm the cytotoxic and/or cytostatic effects induced by the treatment with IR and/or the different senolytics. Microphotographs of cells in representative fields after treatments were taken at 400X magnification using an invertoscope (Axiovert 200; Zeiss, Milan, Italy).

### 4.2. Colony Formation Assay

Colony formation assay was performed as previously described [73]. Briefly, SAOS, HT29, SAOS400, and HT500 cells were seeded at low density (1 × 10^3^ cells/well) into a 35 mm-well plate, irradiated with the indicated doses of γ-rays, and cultured at 37 °C for 2 weeks. This incubation time was carefully selected to avoid clones overlapping and allow an accurate colony count. Subsequently, cells were washed in phosphate-buffered saline (PBS; Euroclone, Milan, Italy), fixed using 10% formaldehyde in PBS, and stained with 0.5% Crystal Violet. Results were expressed as fractions of the number of clones with more than 50 cells with respect to untreated cells. The surviving fraction (S.F.) was calculated as follows: S.F. = colonies counted/(cells seeded x P.E./100), where P.E. represents the plating efficiency, defined as the percentage of seeded cells that formed colonies (comprising > 50 cells) under the specific culture conditions.

### 4.3. Cell Cycle Analysis

After 96 h from IR (5 Gy), SAOS400 and HT500 cells (1.5 and 2 × 10^6^/mL, respectively) were harvested, counted using 0.4% Trypan blue solution, washed in PBS, and fixed in ice-cold 70% ethanol. After two additional washes in cold PBS, cells were stained using 50 μg/mL propidium iodide in the presence of 100 μg/mL DNAse-free RNAase A for 1 h at 37 °C in the dark. Cells were analysed by flow cytometry (Facscalibur; BD Biosciences, Sparks, MD, USA), and DNA content quantified using ModFit LT software (Verity Software House, Inc., Topsham, ME, USA) to measure the percentage of diploid cells distribution in the different phases of the cell cycle [74].

### 4.4. Intracellular ROS and GSH Measurements

Cells (2 × 10^4^/mL) were incubated for 30 min in the presence of the intracellular peroxides probe, chloromethyl derivative of di-chloro-fluorescein diacetate (10 μM CM-DCFDA; Thermo-Fisher Scientific, Molecular Probe, Milan, Italy), or 10 μM superoxide probe dihydroethidium (DHE; Thermo Fisher Scientific, Molecular Probe, Milan, Italy) in PBS at 37 °C in a humidified 5% CO2 atmosphere [75]. Fluorescence was measured with Synergy HT (BioTek, Milan, Italy) multiwell reader with excitation of 485 nm and emission of 530 nm for CM-DCFDA or with excitation of 400 nm and emission of 490 nm for DHE and expressed as DCFH fluorescence (%) or DHE fluorescence (%) with respect to untreated cells.

For GSH intracellular measurement, untreated and treated cells (5 × 10^4^/mL) were incubated for 30 min in PBS with monochlorobimane (50 μM mCB, Merck/Sigma) allowing intracellular GSH S-transferase to form GSH–mCB adducts that were detected fluorimetrically. Finally, cells were washed and fluorescence measured (excitation at 380 nm and emission at 470 nm) in a Synergy HT (BioTek, Milan, Italy) multiwell reader, as described [76]. The results were expressed as a percentage of mCB fluorescence. Each experimental point was repeated three times in quadruplicate.

### 4.5. Senescence Markers

To evaluate senescence markers, we used two different methods: i. senescence β-galactosidase (βGal) staining solution (Merck/Sigma, Milan, Italy), a method based on the histological staining for βGal activity at pH 6.0; ii. a quantitative method based on the alkalinisation of lysosomes with bafilomycin A1 (Enzo Life Science, Milan, Italy) followed by the use of 5-dodecanoylaminofluorescein di-β-D-galactopyranoside (C_12_FDG; Merck/Millipore, Milan, Italy), a fluorogenic substrate for βGal activity [77]. Briefly, cells (at early passages) were seeded at sub-confluent density (1.0–1.5 × 104/well) on 35 mm tissue culture plates. After 72 h from IR, cells were washed with PBS and fixed in 3% formaldehyde/PBS (Merck/Sigma, Milan, Italy) for 5 min at room temperature. Cells were subsequently washed with PBS and stained in fresh senescence βGal staining solution containing 1 mg/mL of 5-bromo-chloro-indolyl β-D galactoside (X-Gal; Merck/Sigma,) in a buffer consisting of 40 mM sodium phosphate, 150 mM NaCl, 5 mM C₆FeK₄N₆, 2 mM MgCl_2_, and pH 6. After overnight incubation at 37 °C, cells were washed with PBS, and microphotographs were taken with an invertoscope (Axiovert 200, Zeiss,). Cells were visually scored as positive or negative for βGal staining. At least 100–200 cells were counted/sample in duplicate and expressed as a percentage of βGal positive cells/total cells.

The C_12_FDG staining protocol was adopted from Debaq-Chainaux et al. [77]. Cells (1.5 × 10^4^ well in quadruplicate) were treated at the indicated time after IR, were combined with bafilomycin A1 (100 nM) for 1 h, followed by incubation with C12FDG (33 μM) for 2 h. Subsequently, cells were washed with PBS and stained with Hoechst 33,342 (Enzo Life Science) for 30 min in a complete DMEM medium. Cellular positivity for senescence markers was obtained by dividing C_12_FDG-derived fluorescence (485/530 nm) with fluorescence derived by nuclear staining with Hoechst 33,342 using a microplate fluorescence reader (Synergy HT, Milan, Italy). Images were obtained using a fluorescence microscope (magnification 400x, Axiovert 200 Zeiss).

### 4.6. Immunoblotting

Cells were lysed using a lysis buffer containing protease and phosphatase inhibitors, as reported [68]. After the measurement of protein concentration (Bradford 1976), the total lysates (20 μg/lane) were loaded on a 4–12% precast gel (Novex Bis-Tris precast gel; Thermo-Fisher Scientific/Life Technologies) using 50 mM MES (2-(Nmorpholino) ethanesulfonic acid) buffer at pH 7. In some experiments, protein lysates (20 μg) were added with 2X Laemmli loading buffer, heated at 95° C for 5 min, and loaded on a 15% SDS–PAGE at pH 8, as previously described [78]. The immunoblots were performed adopting standard procedures using polyvinylidene fluoride (PVDF) membranes that were incubated for about 16 h with the following primary antibodies: anti-p16^INK4^ (code #SC-759), anti-p21^CIP1^ (code #SC-397), both from Santa Cruz Biotechnology (diluted 1:1000 in Tween20-Tris Buffer Salinum, T-TBS, containing 3% Bovine Serum Albumine, BSA), and anti-α-tubulin (code #T9026 Merck/Sigma diluted 1:3000 with 3% BSA in T-TBS). Membranes were washed in T-TBS and finally incubated 2 h with a horseradish peroxidase-linked secondary antibody (diluted 1:20.000 in T-TBS) raised against mouse or rabbit. The immunoblots were developed using the ECL Prime Western blotting detection system kit (GE Healthcare, Milan, Italy). Band intensities were quantified and expressed as optical density on a Gel Doc 2000 Apparatus (Bio-Rad Laboratories, Milan, Italy) and Multianalyst software (Bio-Rad Laboratories).

### 4.7. Combination Index Assessment

The combination index (C.I.) was calculated according to the Chou and Talalay mathematical model for drug interactions as previously reported [43,79]. The dose-response curves were based on the evaluation of the fraction affected (f.a), indicating the percentage of dead cells with respect to the untreated. Dose-effect analysis and C.I. for the combined treatments groups (quercetin or ABT-263 plus IR at the indicated doses) were generated using the CompuSyn software (freely available at www.combosyn.com; accessed on 8 July 2021).

### 4.8. Statistical Analysis

Statistical Analysis: IBM SPSS Statistics (Version 23.0. IBM Corp., Armonk, NY, USA) was used for the statistical analyses, and statistical significance was accepted at *p*-value less than 0.05. Results have been expressed as mean ± standard deviation (SD) based on values obtained from independent experiments performed in triplicate or quadruplicate. Differences between two groups were analyzed using the Students’ *t*-test, and specific values were indicated in figure legends. For multiple group comparisons at a single time point, one-way ANOVA with Bonferroni’s multiple comparisons test was used (Figure 6a, Figure 7a and Figure 8a–d, and Appendix A). We performed repeated measures ANOVA for multiple comparisons at different time points (Figure 5a–d and Appendix A).

## Figures and Tables

**Figure 1 ijms-23-00301-f001:**
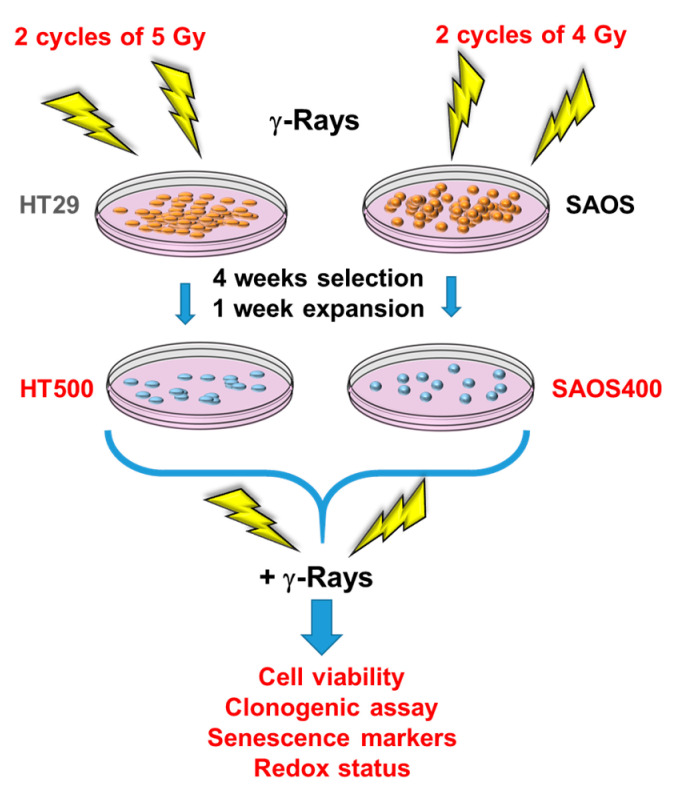
Scheme summarizing the procedures applied to obtain radio-resistant HT500 and SAOS400 cell lines from their parental HT29 and SAOS cells, respectively.

**Figure 2 ijms-23-00301-f002:**
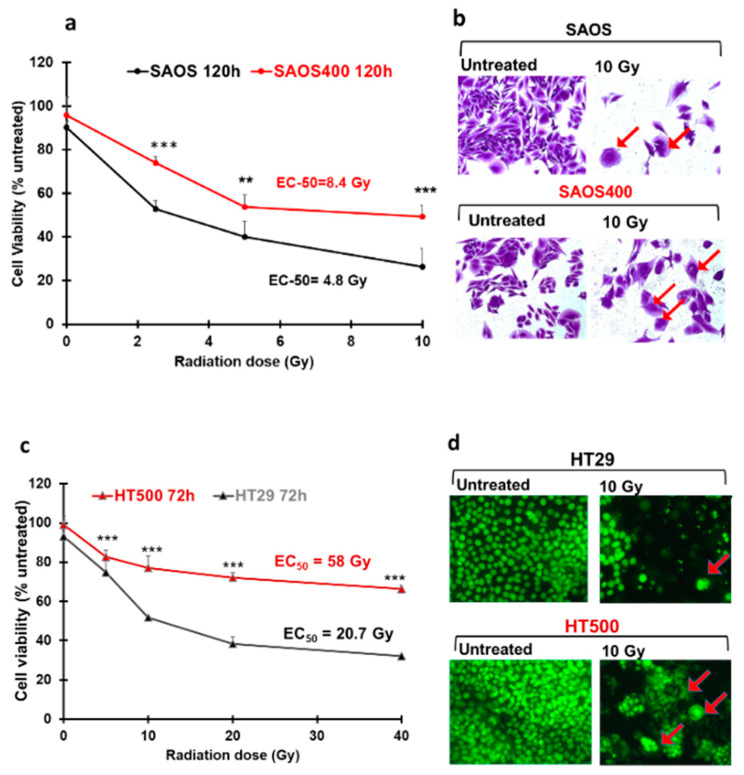
Radio-resistance of SAOS400 and HT500 cells compared to their parental cell lines. Changes in cell viability after increasing doses of γ-rays in SAOS vs. SAOS400 (panel **a**) and HT29 vs. HT500 cells (panel **c**). Data represent the mean of three independent experiments (±SD). Symbols indicate significance: ** *p* < 0.01 or *** *p* < 0.001 of the parental cells compared to the radio-resistant sub-population (Student’s *t*-test). The calculated EC50s are reported as inserts within the graphs. Photographs (400x magnification) show representative fields of untreated and SAOS or SAOS400 cells after IR (10 Gy), stained with Crystal Violet dye (panel **b**) or untreated and IR-treated (10 Gy) HT29 or HT500 cells stained with CyQuant dye (panel **d**).

**Figure 3 ijms-23-00301-f003:**
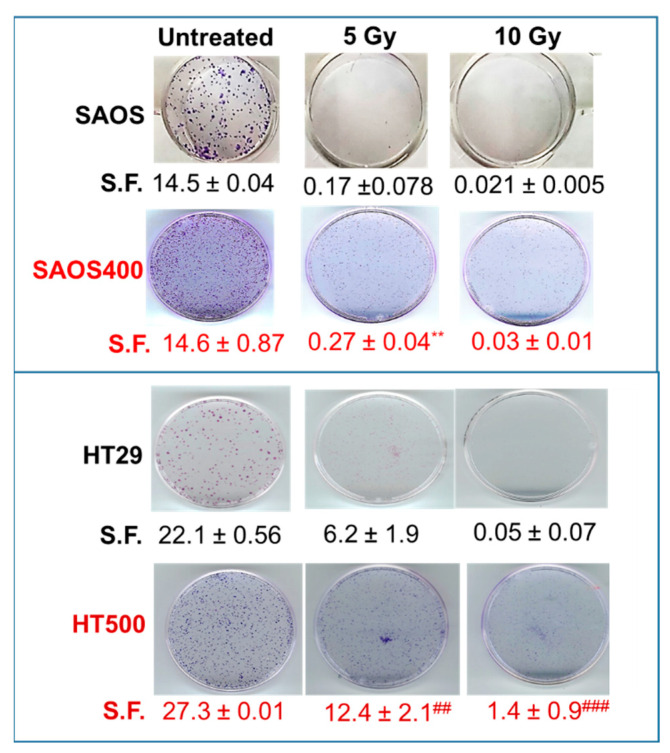
Colony-forming assay. Representative images are reported comparing SAOS and HT29 cells with their respective radio-resistant counterparts, SAOS400 and HT500 cells. Numbers below each row of panels indicate the means of counted colonies ± SD expressed as S.F (Surviving Fractions) from two independent experiments. Symbols indicate significance with ** *p* < 0.01 with respect to SAOS cells and ^##^ *p* < 0.01 and ^###^ *p* < 0.001 with respect to HT29 cells (Student’s *t*-test).

**Figure 4 ijms-23-00301-f004:**
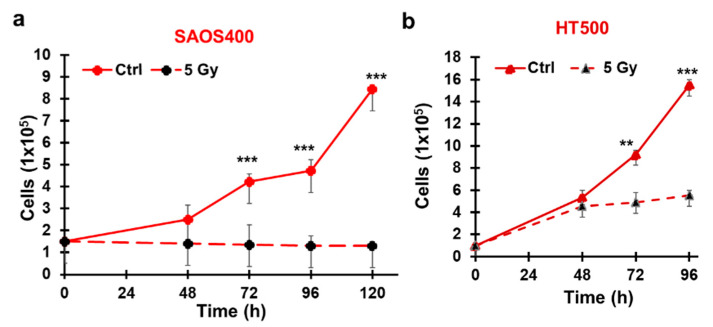
Effect on cell growth after irradiation in SAOS400 and HT500 cell lines. The number of viable cells after 5 Gy irradiation in SAOS400 (panel **a**) and HT500 (panel **b**) cells was evaluated at the indicated time using Trypan blue exclusion dye. Data reported indicate the mean ± SD of two independent experiments; symbols indicate significance: ** *p* < 0.01 and *** *p* < 0.001 with respect to IR cells. (Panels **c**,**d**) report the cell cycle analyses of SAOS400 and HT500 cells, respectively after γ−rays irradiation (5 Gy). Cells were irradiated, harvested after 96 h, fixed, stained with propidium iodide, and analysed by flow cytometry as described in the Methods section. The left panels report representative histograms of cell cycle distribution obtained using ModFit software, while in the insert tables on the right, the percentages of diploid cells in the different phases of the cell cycle are indicated.

**Figure 5 ijms-23-00301-f005:**
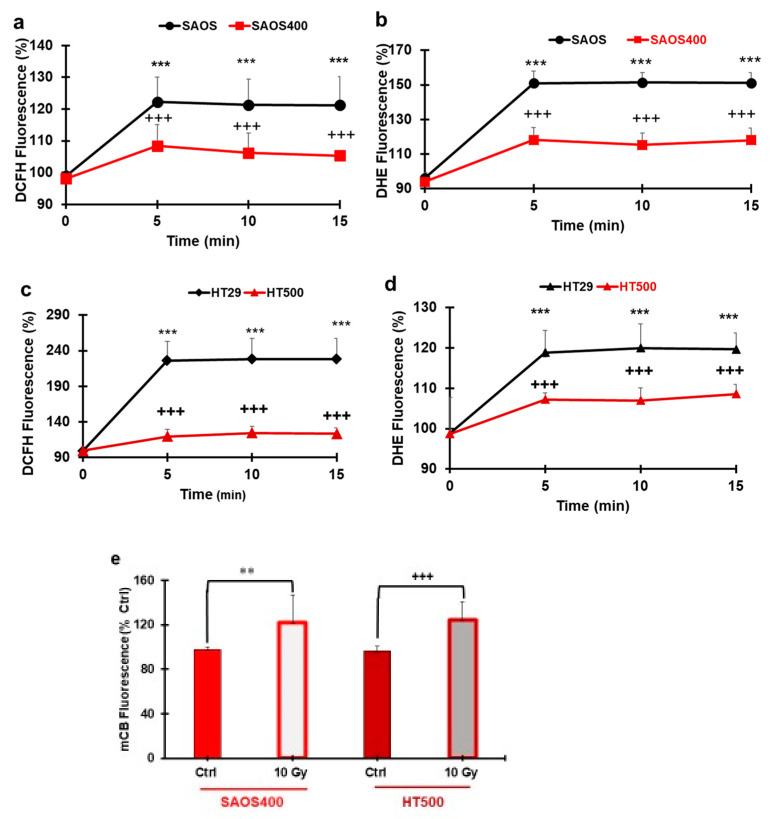
ROS and GSH production in SAOS400 and HT500 cells. SAOS/SAOS400 and HT29/HT500 (panels **a** and **c**) cells were pre-incubated with H_2_O_2_ probe (CM-DCFDA) or O_2_^−^ probe (DHE) (panels **b** and **d**), washed, and irradiated (10 Gy). Fluorescence was measured in the 5–15 min time range. Data reported are the mean ± SD of three independent experiments in quadruplicate. Symbols indicate significance after ANOVA analysis for multiple comparisons at different time points: *** *p* ≤ 0.001 of SAOS400 and HT500 cells vs. their parental cell lines; +++ *p* ≤ 0.001, indicate significance of SAOS400 and HT500 cells at times 5–15 min vs. time 0 min. In panel (**e**), the GSH levels in SAOS400 and HT500 cells after 10 Gy for 2 h are reported. Cells were incubated with a GSH intracellular probe (mCB), as described in the Methods section. Bar graphs represent the mean ± SD of three independent experiments in quadruplicate. Symbols indicate significance: ** *p* < 0.01 untreated SAOS400 cells vs. irradiated; +++ *p* < 0.001 untreated HT500 vs. irradiated.

**Figure 6 ijms-23-00301-f006:**
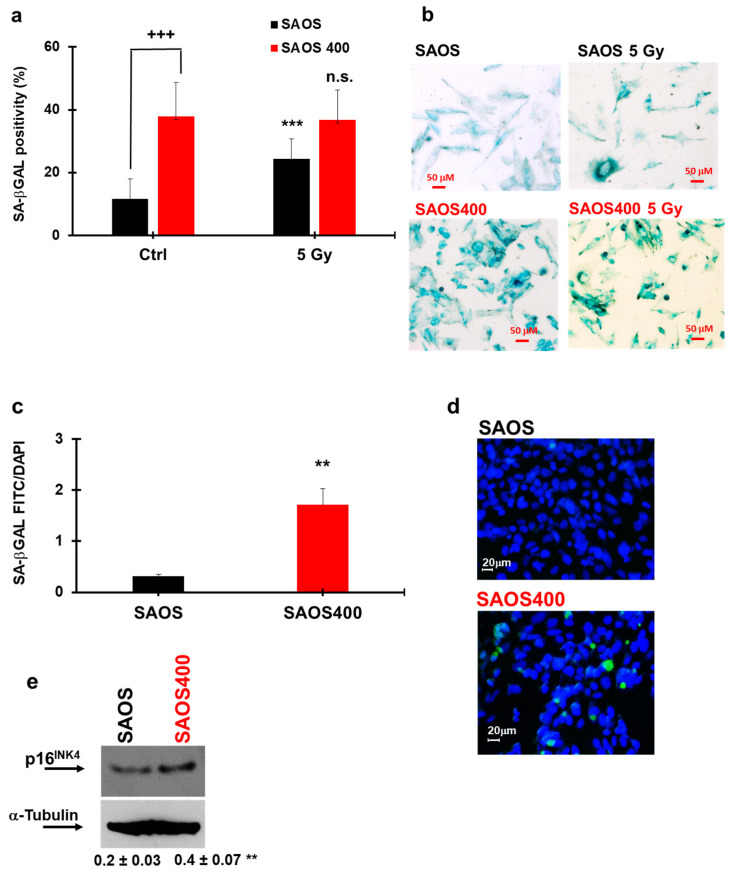
Senescence measurements in SAOS and SAOS400 cell lines. SA-βGal staining in SAOS and SAOS400 cells before and after irradiation (5 Gy) were quantified (panel **a**) and microscopically visualized (panel **b**). The results of a different protocol that employed C_12_FDG staining were also quantified (panel **c**), and representative images were reproduced (**d**). Bar graphs represent the mean ± SD of two independent experiments. Symbols indicate significance: +++ *p* < 0.001 for untreated SAOS vs. untreated SAOS400 cells; *** *p* < 0.001 for untreated SAOS vs. irradiated SAOS (black bars); n.s. not significant in SAOS400 before and after irradiation (red bars) One-way ANOVA with Bonferroni’s multiple comparisons test was used. Panel (**e**) reports the immunoblot showing the basal expression of p16^INK4^ in SAOS and SAOS400 cells. Numbers between panels indicate the densitometric analysis ± SD of three independent experiments. Symbols indicate significance: ** *p* < 0.01 vs. SAOS (T-Test Student).

**Figure 7 ijms-23-00301-f007:**
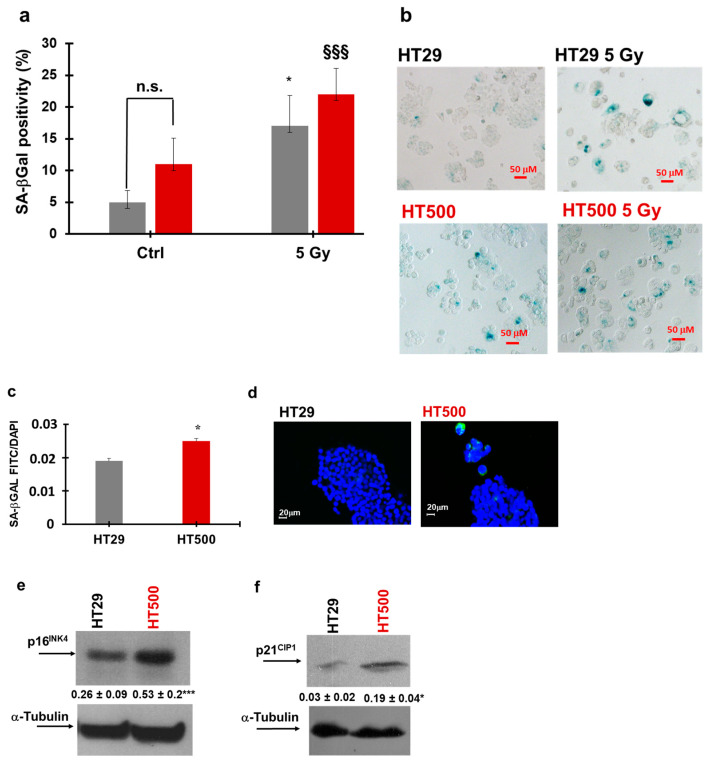
Senescence measurements in HT29 and HT500 cell lines. SA-βGal staining in HT29 and HT500 cells before and after irradiation (5 Gy) were quantified (panel **a**) and microscopically visualized (panel **b**). The results of a different protocol that employed C_12_FDG staining were also quantified (panel **c**), and representative images were reproduced (**d**). Bar graphs represent the mean ± SD of two independent experiments. Symbols indicate significance: * *p* < 0.05 for untreated HT29 vs. irradiated HT29 (grey bars); §§§ *p* ≤ 0.001 for HT500 before and after irradiation (dark red bars) One-way ANOVA with Bonferroni’s multiple comparisons test was used. The immunoblots report the basal expression of p16^INK4^ (panel **e**) and p21^CIP1^ (panel **f**) in HT29 and HT500 cells. Numbers between panels indicate densitometric analysis ± SD of two independent experiments. Symbols indicate significance with * *p* < 0.05 and *** *p* < 0.001 (*t*-test Student).

**Figure 8 ijms-23-00301-f008:**
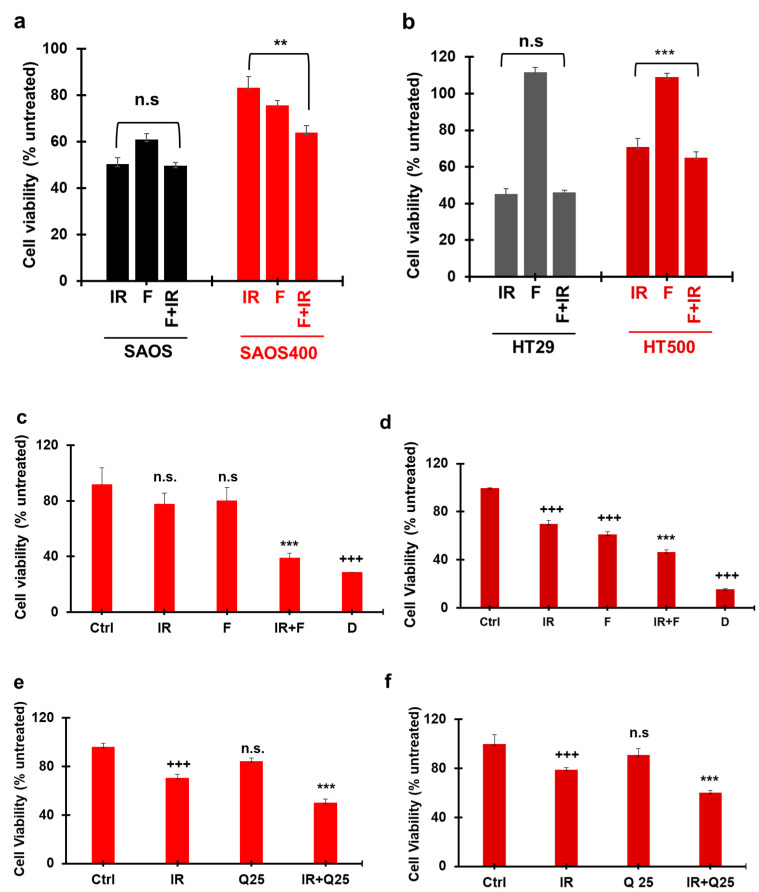
Senolytic effects of fisetin and quercetin in irradiated SAOS400 and HT500 cell lines. SAOS and SAOS400 cells (panel **a**) and HT29 and HT500 cells (panel **b**) were pre-irradiated (10 Gy), cultured for 72 h, and subsequently were incubated for an additional 72 h in the presence of 20 μM F. CyQuant assay was used to quantify cell viability expressed as a percentage of untreated cells. Bar graphs represent mean ± S.D of two independent experiments. Symbols indicate significance: *p* ≤ 0.01 ** *p* ≤ 0.001 *** with respect to F and IR cells. n.s. = no statistical significance between the combined treatment, F + IR, vs. IR mono-treatment in both SAOS and HT29 parental cell lines. Radio-resistant SAOS400 (panel **c**) and HT500 (panel **d**) cells were pre-irradiated and then directly treated for 96 h with 40 μM F. CyQuant assay was used to quantify cell viability expressed as a percentage of untreated cells. Bar graphs represent mean ± S.D of two independent experiments. Symbols indicate significance: +++ *p* ≤ 0.001 vs. untreated (Ctrl); *** *p* ≤ 0.001 vs. F and IR; n.s. not significant vs. untreated (Ctrl). Daunorubicin (D; 0.04 mg/mL) was used as positive control. In panel (**e**) and (**f**), SAOS400 and HT500 cells were pre-irradiated (5 Gy) and cultured for 72 h; subsequently, they were incubated for an additional 48 h with 25 μM Q or vehicle DMSO (0.1%). Crystal Violet assay was used to quantify cell viability that was expressed as a percentage of DMSO-treated cells. The bar graph represents mean ± S.D. Symbols indicate significance: +++ *p* ≤ 0.01 vs. untreated, *** *p* < 0.001 vs. Q/IR; n.s. not significant vs. untreated (Ctrl). One-way ANOVA with Bonferroni’s multiple comparisons test was used in these experiments.

**Figure 9 ijms-23-00301-f009:**
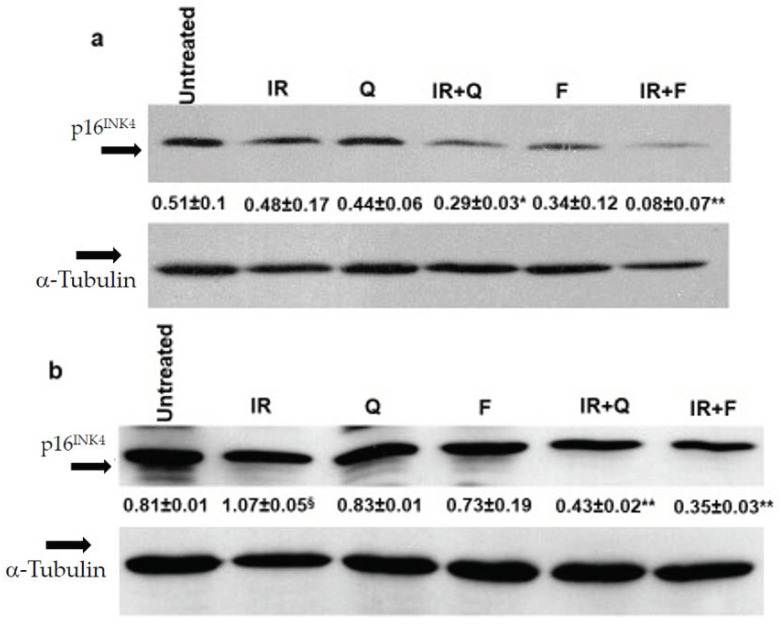
Changes in senescence markers following senolytic treatments in SAOS400 and HT500 cell lines. Immunoblot analysis of p16^INK4^ expression in SAOS400 cells (panel **a**) pre-irradiated (10 Gy) and treated with F (40 μM) or Q (40 μM) for 48 h. Densitometric analysis (numbers between panels) was obtained normalizing the expression of p16^INK4^ with α−tubulin and quantified as described in the Methods section. * *p* < 0.05 vs. Q and IR; ** *p* < 0.01 vs. F and IR. Immunoblots of p16^INK4^ (panel **b**) and p21^CIP1^ (panel **c**) expressions in HT500 cells pre-irradiated (10 Gy) and treated with 40 μM F or 40 μM Q for 72 h. Densitometric analyses (numbers between panels) were obtained normalizing the expressions of p21^CIP1^ and p16^INK4^ with α-tubulin and quantified as described Methods section. § *p* < 0.05 vs. untreated; §§ *p* < 0.01 vs. untreated; * *p* < 0.05, ** *p* < 0.01 vs. F plus Q and IR.

**Figure 10 ijms-23-00301-f010:**
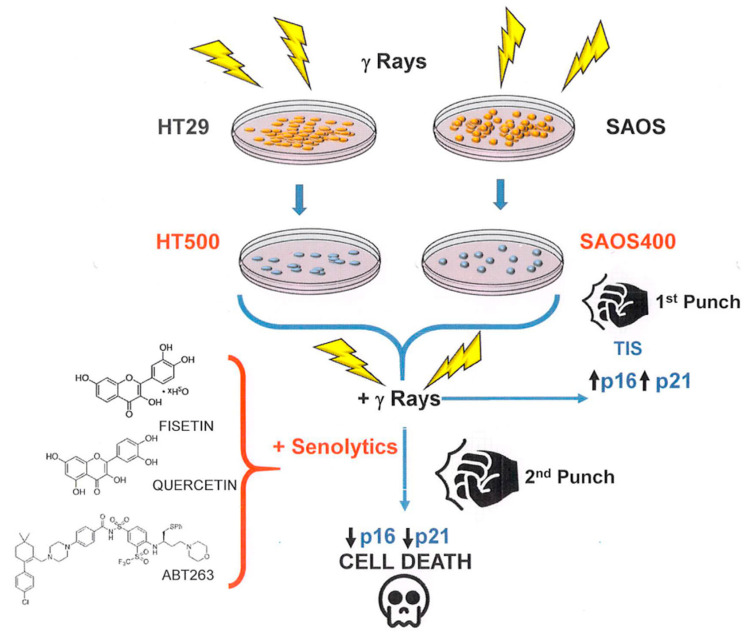
General cartoon showing how TIS can play the double role of a form of resistance to cell death but also as a process of “therapeutic vulnerability” for senolytic drugs.

**Table 1 ijms-23-00301-t001:** SAOS400.

Quercetin (μM)	IR (Gy)	Effect ^§^	C.I. *
SAOS400			
25	5	0.461	0.75
50	5	0.628	0.69
HT500			
25	5	0.4	0.91
50	5	0.63	0.85
SAOS400			
ABT-263 (μM)	IR (Gy)	Effect	C.I. *
0.5	5	0.24	0.71
1.0	5	0.51	0.50
HT500			
0.5	5	0.39	0.36
1.0	5	0.42	0.37

Calculation of the combination index in irradiated SAOS400 and HT500 cells in the presence of quercetin and ABT-263. ***** C.I. for the combination treatment groups were generated using the equations reported by Chou and Talalay [43] using the CompuSyn software based on fraction affected (§: f.a.; “Effect” in Table 1) obtained with different doses of IR and molecules at a not-constant ratio, as described.

## Data Availability

Not applicable.

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
