# Peer review of "Biochemical and Cellular Characterization of New Radio-Resistant Cell Lines Reveals a Role of Natural Flavonoids to Bypass Senescence"

_ijms, 2021, doi:10.3390/ijms23010301_

Round 1

Reviewer 1 Report

Dear authors,

This is an interesting article in terms of methodology. Radio-resistance is an important issue in cancer. The creation of cell lines to simulate this effect and try to look for potential treatments to overcome this resistance is imperative to help cancer patients. I have few comments regarding the statistics:

-Figure 5. The significant symbols are missing in the legend.

  • In order to use a T-test two conditions must be complied: variance homogeneity and normal distribution. Have you checked this in your data? If not, please, check. If these conditions fail, then you have to apply no-parametric tests. Also, T-test can only be applied when you compare two groups. If you compare three groups, you have to use the ANOVA.

Thank you very much. 

Author Response

Reviewer 1

This is an interesting article in terms of methodology. Radio-resistance is an important issue in cancer. The creation of cell lines to simulate this effect and try to look for potential treatments to overcome this resistance is imperative to help cancer patients.

We thank Reviewer-1 for the very positive opinion expressed on our manuscript and for considering it highly significant.

Query-1. I have few comments regarding the statistics.

Figure 5. The significant symbols are missing in the legend.

Reply-1. Figure 5, as others, have been largely revised according with Reviewer-1’s requests (see aslo comments below).

Query-2. •      In order to use a T-test two conditions must be complied: variance homogeneity and normal distribution. Have you checked this in your data? If not, please, check. If these conditions fail, then you have to apply no-parametric tests. Also, T-test can only be applied when you compare two groups. If you compare three groups, you have to use the ANOVA.

Reply-2. We thank the Reviewer-1 for the keen comment. We accordingly largely revised the Statistical Analysis. As for the statistical tests, we performed both parametric and non-parametric tests and results were basically overlapping. However, in order to be consistent with literature we have decided to show results of parametric test. Moreover, following your suggestion, for multiple group comparisons at a single time point “one-way ANOVA” (with Bonferroni’s multiple comparisons test) was used. In the case of multiple comparisons at different time points, we performed “repeated-measures ANOVA” (with Bonferroni’s multiple comparisons test).

All changes are reported in the figures and respective legends. The statistical values have been uploaded as supplementary data (Table S1). A new paragraph has been also added in the Materials and Methods section (pag. 20 of the revised manuscript).

Reviewer 2 Report

The work is really interesting but I have some comments:

Comment 1: In the Figure 5, the authors should include the legend

Comment 2: In the Figure 6b, 6e, 6d, 7b, 7d, the authors should include a scale bar

Comment 3: The Figure 6e, the authors should provide a better WB

Author Response

Query-1. In the Figure 5, the authors should include the legend.

Reply-1. The legend is now present in Figure 5, where an upgraded of the statistical analysis has been performed following Reviewer-1 requests.

Query-2. In the Figure 6b, 6e, 6d, 7b, 7d, the authors should include a scale bar.

Reply-2. Scale bars have been added in all figures as requested.

Query-3. The Figure 6e, the authors should provide a better WB.

Reply-3. We selected an immunoblotting of better quality for Fig. 6e and replaced the original one.
